# Metallogenesis of Porphyry Copper Deposit Indicated by In Situ Zircon U-Pb-Hf-O and Apatite Sr Isotopes

Hong Zhang [1],*, Fang An [1], Mingxing Ling [2], Xiaolin Feng [1] and Weidong Sun [3]

1 State Key Laboratory of Continental Dynamics, Department of Geology, Northwest University, Xi'an 710069, China
2 State Key Laboratory of Nuclear Resources and Environment, East China University of Technology, Nanchang 330013, China
3 Center of Deep Sea Research, Institute of Oceanology, Chinese Academy of Sciences, Qingdao 266071, China
* Correspondence: zhanghong820426@163.com

**Abstract:** The origin of the Dexing porphyry Cu deposit is hotly debated. Zircon and apatite are important accessory minerals that record key information of mineralization processes. SHRIMP zircon U-Pb analyses of granodiorite porphyries yield ages of $168.9 \pm 1.2$ Ma, $168.0 \pm 1.0$ Ma, and $172.8 \pm 1.3$ Ma, whereas zircons in the volcanic rocks of the Shuangqiaoshan Group have Neoproterozoic ages of $830 \pm 7$ Ma, $829 \pm 8$ Ma, and $899 \pm 12$ Ma. The porphyry displays zircon in situ $\delta^{18}O$ of mantle values ($5.5 \pm 0.2‰$), low apatite $^{87}Sr/^{86}Sr$ ratios ($0.7058 \pm 0.0005$), and high $\varepsilon_{Hf}$ values ($5.1 \pm 1.5$), which are consistent with mantle-derived magmatic rocks. Apatite from the porphyries has relatively high total rare earth elements (REEs) and negative Eu anomalies, with relatively high Cl and As contents. These features are distinctly different from apatite in the Shuangqiaoshan Group, which shows lower total REE, Cl, and As contents but higher F content and positive Eu anomalies. Zircon in porphyries yields a relative high oxygen fugacity of $\Delta FMQ + 1.5$ based on zircon $Ce^{4+}/Ce^{3+}$. Apatite in porphyries also shows high oxygen fugacity based on its $SO_3$ and Mn compositions, reaching $\Delta FMQ + 2$, which is different from that of the lower continental crust in general, but similar to subduction-related magmas. In contrast, the oxygen fugacity of the Shuangqiaoshan Group is much lower, suggesting a different origin for its wall rock. Therefore, the Dexing porphyries were not derived from the lower crust but derived from partial melting of the subducting Paleo-Pacific plate.

**Keywords:** porphyry copper deposit; metallogenesis; in situ zircon U-Pb-Hf-O isotopes; in situ apatite Sr isotopes; ore-forming fluids

## 1. Introduction

Porphyry deposits are significant repositories of copper, gold, and molybdenum, which are characterized by low-grade copper, gold, and/or molybdenum mineralization developed within and around a porphyritic intrusive complex. Most of the known large-scale porphyry copper deposits in the world are distributed along island arc and continental margin arc environments [1–3]. They are characterized by high oxygen fugacity [2–6]. However, the host rock of Cu porphyry deposits is generally highly altered, such that the original information is erased. Therefore, it remains controversial whether porphyry Cu deposits are associated with partial melting of subducted oceanic crust [3,7], mantle wedge [8,9], or juvenile lower crust [10–12].

Some accessory minerals, e.g., zircon and apatite, are stable during later hydrothermal alterations. Zircon records the composition and primary redox state of magma. In contrast, apatite crystallizes in both magma and hydrothermal fluids and, thus, is sensitive in recording the compositional and physicochemical evolution of a combined magma–hydrothermal system [13–15]. Zircon $Ce^{4+}/Ce^{3+}$, S, and Mn in apatite are acknowledged as reliable redox proxies [3,5,16–20]. Therefore, the compositions of zircon and apatite can be used to constrain the origin and evolution of magmas associated with porphyry Cu-Mo-Au

mineralization (e.g., Pizarro et al. [15]). Meanwhile, apatite occurs more commonly than zircon in different types of rocks and hydrothermal deposits as an important carrier of U, Th, Sr, rare earth element (REE), and volatile elements (F, Cl, OH, and S), and is widely adopted to trace petrogenesis, magmatic evolution process, magmatic oxidation state, and hydrothermal process [13,21–31].

The Dexing Cu deposit is the largest porphyry Cu deposit in southeast China [32,33]. The origin of this deposit has been highly debated. Some researchers suggest that the deposit is a typical intra-plate deposit with mineral sources from the Neoproterozoic volcanic sequences [11,34–36], while others propose that metallogenesis was associated with the subducting Paleo-Pacific plate [7,37–39].

In this study, we conducted in situ U-Pb-Hf-O and Sr isotope analyses of zircon and apatite from the porphyries and wall rocks (Shuangqiaoshan Group) of the Dexing porphyry deposit. The new datasets, combined with data in previous studies, are utilized to constrain the origin and evolution of ore-forming magma and hydrothermal systems. Our results provide strong lines of evidence to support the conclusion that the parental magma of the deposit is highly oxidized and the metallogenesis is associated with the Jurassic subduction of the Paleo-Pacific plate.

## 2. Geological Background

### 2.1. Regional Geology

The Dexing porphyry Cu deposit in southeast China is located within the interior of the South China Block, which is made up of two lithospheric blocks, the Yangtze and Cathaysia blocks (Figure 1a). The Dexing district is located along the southern margin of the Yangtze block, bounded by the Jiang-Shao Fault, a Neoproterozoic suture zone. To the north of the Jiang-Shao Fault, the Precambrian sedimentary basement, consisting of metamorphic Neoproterozoic strata, is widely exposed in the Dexing district [40–42]. To the south of the Jiangshao fault, the sedimentary basin is developed along the northern margin of the Cathaysia block [33,43]. The NE-trending Gan-Hang rift zone is approximately parallel to the Jiangshao Fault and is marked by a series of NE-trending Jurassic-Cretaceous basins and numerous intrusions of A-type granites, gabbros, and associated basalts (Figure 1b,c) (183–152 Ma; [12,44,45]). Geophysical data indicate that this fault is translithospheric, and offsets the lower crust by several kilometers [33]. The E-W- and NNE-trending faults associated with the Jiang-Shao Fault have a long history of activity during the mid-Jurassic time: some of the older faults control the location of the Dexing porphyry Cu systems, whereas some of the younger faults cut the porphyry stocks in the Dexing area. There are a series of NE-trending Jurassic to Cretaceous extensional basins and abundant A-type granites in the neighborhood of the Dexing region [11,46]. The genesis of these copper-gold deposits is closely related to the adakitic intrusive rocks.

### 2.2. Deposit Geology

The Dexing porphyry Cu deposit consists of three mines: Tongchang mine in the center, Fujiawu mine in the southeast, and Zhushahong mine in the northwest [33,47] (Figure 1c). The main metallogenic type is granodiorite porphyry with adakite characteristics, such as high $Al_2O_3$ and Sr contents, high La/Yb and Sr/Y ratios, and low Yb and Y contents, where the crystallization and metallogenic ages are generally concentrated in the range of 168~173 Ma [11,35,37,38,46–51]. The Tongchang mine, with an exposed area of the Tongchang porphyry of about 0.7 km$^2$, contains 5 Mt Cu, 110,100 t Mo, and 188 t Au. The Fujiawu mine reserves are 2.506 Mt of Cu and 163,100 t of Mo. The Zhushahong mine has Cu reserves of only 60,800 t. The corresponding grades of copper, molybdenum, and gold are 0.47%, 0.01% and 0.18 g/t, respectively, which are the most typical super-large porphyry Cu-Mo-Au deposits in South China. Granitic porphyries in open pits can be found in the Tongchang district, different from the intensely altered Zhushahong and Fujiawu deposits, where some porphyry can only be picked out from drill holes (Figure 2a,b).

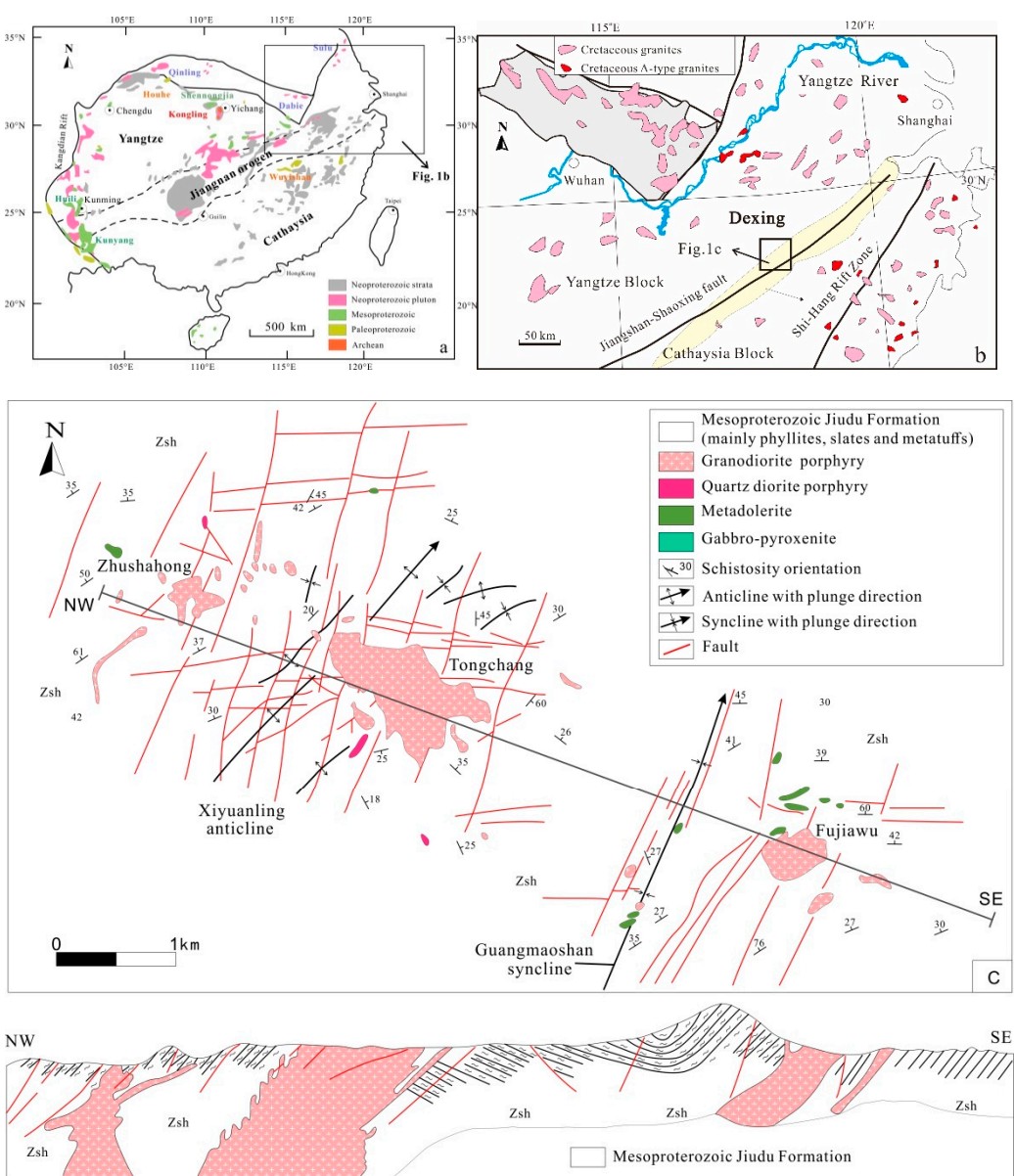

**Figure 1.** (**a**) Simplified geologic map of the South China Craton (after Zheng et al. [52]), showing the distribution of Precambrian rocks containing the Shuangqiaoshan Group; (**b**) geologic map of southeast China, showing the main granitic porphyries and location of the Dexing porphyry Cu (-Mo-Au) district; (**c**) geologic map and longitudinal section from the Dexing deposit (after Yang et al. [53]).

The main rock types in the mineralized porphyries are light grey or pink porphyritic granodiorites, characterized by 40–60 vol.% phenocrysts, mainly consisting of euhedral-subhedral plagioclase (0.4–3 mm; 40–40 vol.% andesine) and subordinate euhedral-subhedral K-feldspar (0.5–2.0mm; 10–20 vol.%), hornblende (0.5–3.0 mm; 5–10 vol.%), biotite, and minor round grains of quartz. The matrix has a microcrystalline (0.05–0.50 mm) texture and consists of hypidiomorphic oligoclase, hornblende, biotite, quartz, and K-feldspar, with minor accessory minerals including magnetite, apatite, titanite, and zircon [34,46].

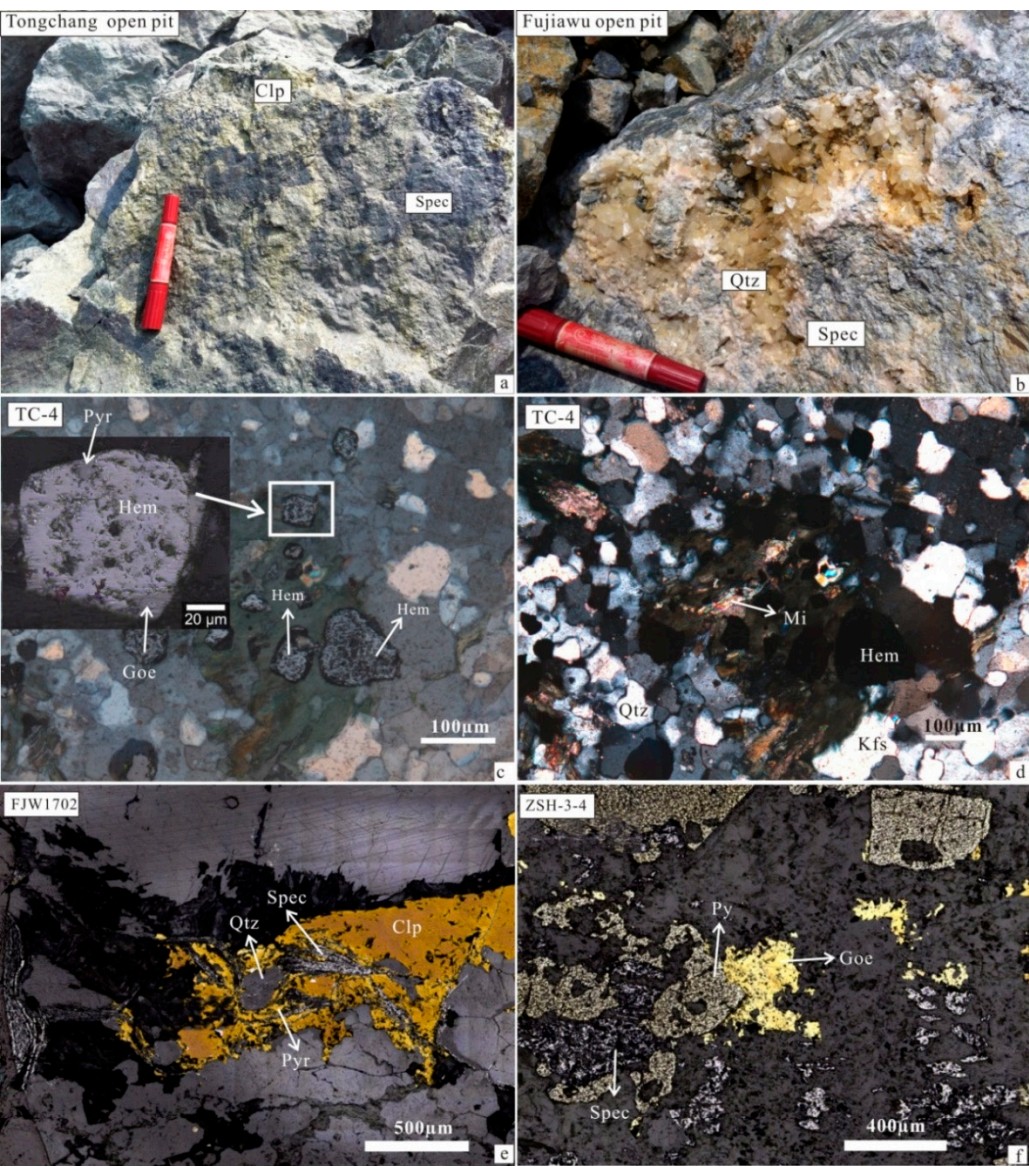

**Figure 2.** Representative porphyry and mineral combination types of Dexing porphyry copper deposit. (**a**) Specularite and chalcopyrite in the Tongchang porphyry; (**b**) quartz vein and specularite in Fujiawu open pit; (**c**) reflected light pictures showing sulfides and oxides in the Tongchang porphyry; (**d**) transmission pictures of different minerals in the Tongchang porphyry; (**e**) sulfide veins from Fujiawu drill hole consists of Qtz, Spec, Clp, and Pyr; (**f**) the sulfide and oxides combination contains Py, Goe, and Spec from Zhushahong drill hole (Qtz, quartz; Clp, chalcopyrite; Goe, goethite; Hem, hematite; Kfs, K-feldspar; Mi, Muscovite; Py, pyrite; Pyr, pyrrhotite; Spec, specularite).

The basement rocks exposed in the Dexing area are greenschist facies volcano-sedimentary strata of the Shuangqiaoshan Group. They include sericite phyllites, slates, metatuffs, and metasandstones, with a total thickness of ≥12,000 m [33].

## 3. Petrography and Alteration

Ore minerals in the Dexing Cu deposit include chalcopyrite, pyrite, bornite, molybdenite, pyrrhotite, tennatite, etc. In addition, there are large amounts of iron oxides such as magnetite; hematite and its variety, specularite [7]; and goethite. Among these, the association of hematite and magnetite represents the high oxygen fugacity environment of deposit formation (Figure 2b–e).

The Dexing porphyry has undergone extensive alteration. Potassic, silicic, and phyllic alteration are all exposed in the open pit and drill core. Potassic alteration in the Dexing district is characterized by K-feldspar, biotite, quartz, magnetite, and anhydrite. Pan et al. [54] and Yang et al. [53] have revealed that A-type quartz and quartz-magnetite veins are widely developed in the Tongchang granodiorite porphyry and in adjacent rocks of the Jiudu Formation, indicating that potassic alteration at Tongchang is better developed than previously thought (e.g., Zhu et al. [33]). Irregular quartz, quartz ± magnetite/specularite, quartz ± K-feldspar, and biotite ± sericite veins are related to potassic alteration at Tongchang mine (Figure 2). The earliest quartz veins hosted along schistosity in rocks of the Jiudu Formation are typically irregular and vary from 0.5 to 3 cm in width. These veins are filled in a large number of the pyrite and chalcopyrite (<1 cm), with minor quartz-magnetite, quartz ± biotite, and quartz ± K-feldspar.

Phyllic alteration is extensively developed in both the granodiorite porphyry intrusions and wall rocks. It is divided into three zones: intensive phyllic alteration, moderate phyllic alteration, and weak phyllic alteration. Phyllic alteration is strongly developed in both the granodiorite porphyry intrusions and rocks of the Jiudu Formation within ~500 m of the mineralizing porphyry intrusions. The intensity of phyllic alteration decreases away from the contact zone. Phyllic alteration can be subdivided into an inner quartz-sericite zone and an outer chlorite-sericite zone. The quartz-sericite zone is characterized by textural destruction of the granodiorite intrusions and metamorphic country rocks, with a well-developed quartz, sericite, and pyrite assemblage [33]. Strong alteration also characterizes the chlorite-sericite zone, with chlorite, sericite, quartz, and pyrite constituting most of the altered rock.

Minor relict epidote observed in the zone is part of the propylitic assemblage. Phyllic alteration is coeval with chalcopyrite-pyrite, chalcopyrite, pyrite, pyrite-chalcopyrite-carbonate, and quartz-chlorite-chalcopyrite veins. Propylitic alteration, characterized by chlorite, epidote, and minor albite, anhydrite, and carbonate, occurs widely in rocks of the Jiudu Formation.

## 4. Sampling and Analytical Methods

### 4.1. Sampling

Six samples, including TC-W-1, TC-12, and TC-13 from the Tongchang open pit and ZSH-3-4-12, FJW-13, and FJW1702-16 from drill cores of the Zhuashahong and the Fujiawu deposits (Table 1), were selected to separate zircon and apatite. Samples from Zhushahong and Fujiawu are siliceous slate with weak phyllic alteration from the Shuangqiaoshan Group, and other samples are porphyry (Table 1).

**Table 1.** Six samples detail information and features of zircon and apatite grains under microscope.

| Number | Sample Location | Rock Types | Minerals' Assemblages | Zircon Features | Apatite Features |
|---|---|---|---|---|---|
| TC-W-1, TC-12, TC-13 | Tongchang open pit | Mineralized porphyry | Pla, Qtz, Kfs, Bi, Amp; Mag, Spe, Ap, Tit, Zr, Goe, Ill, Chl, and Ru; with potassic to siliceous veins | Length~100–400 μm; width~100–300 μm, zoning crystallization, light yellow, magmatic zircons | Length~200–1000 μm; width~50–200 μm crystallization, flesh-red or light brown, scare inclusions, magmatic-hydrothermal apatite |
| ZSH-3-4-12, FJW1702-16, FJW-13 | Zhushahong drilling hole | Mineralized siltstone or phyllite | Ser, Pla, Chl, Qtz; Ap, Zr, Ru, Ilm, Mag, and clay minerals; weak propylitic to phyllic alteration | Length~100–400 μm; width~100–300 μm, zoning crystallization, light yellow, magmatic zircons | ~50–200 μm irregular enriched inclusions colorless to gray-white hydrothermal apatite |

Zircon and apatite grains were separated by traditional method including fine crushing, water flotation, gravity separation, and magnetic separation and then were purified

under binocular magnification. Among them, the zircon grains of porphyry TC-W-1, TC-12, and TC-13 are characterized by large numbers, large grains (about 100–400 µm in diameter), good crystal shape and light yellow appearance. The cathodoluminescence (CL) images of zircon presents typical oscillating zoning (Figure 3). Zircons in the wall rocks ZSH-3-4-12, FJW1702-16, and FJW-13 are characterized by fragmentation, relatively small particle sizes (about 50–200 µm in diameter), enriched inclusions, and lack of obvious zoning development (Figure 3).

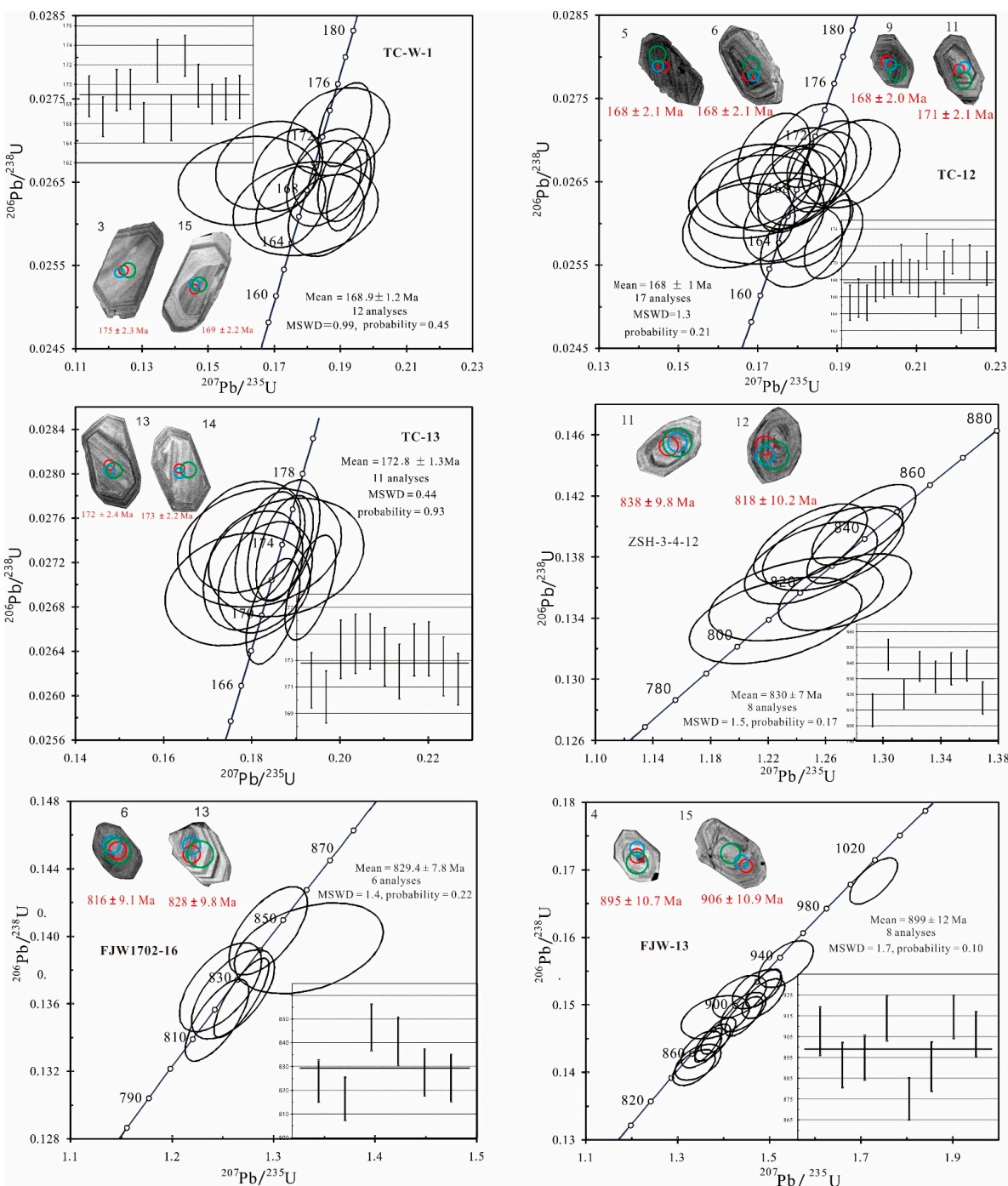

**Figure 3.** Zircon CL images and SHRIMP U-Pb age distribution from the Dexing copper deposit.

Apatite is relatively enriched in the porphyry and wall rocks as an accessory mineral with two different crystallization features (Figure 4). The first type in porphyry shows

typical crystal zoning as magmatic apatite, generally 200~1000-μm long and 50~200-μm wide, with flesh-red or light-brown euhedral columnar crystals. These features of apatite from porphyry TC-W-1, TC-12, and TC-13 are characterized by large size and magmatic-hydrothermal characteristics filled by the later hydrothermal fluid. The second type, from the Zhuashahong and the Fujiawu drill cores, is characterized by fragmentation, relatively small sizes, and inclusions' enrichment (Figure 4). These grains display colorless to gray-white fine needle-like crystals that are anhedral. The particle size generally varies from 50 to 200 μm and exists in the different stage sulfide veins (Figure 4).

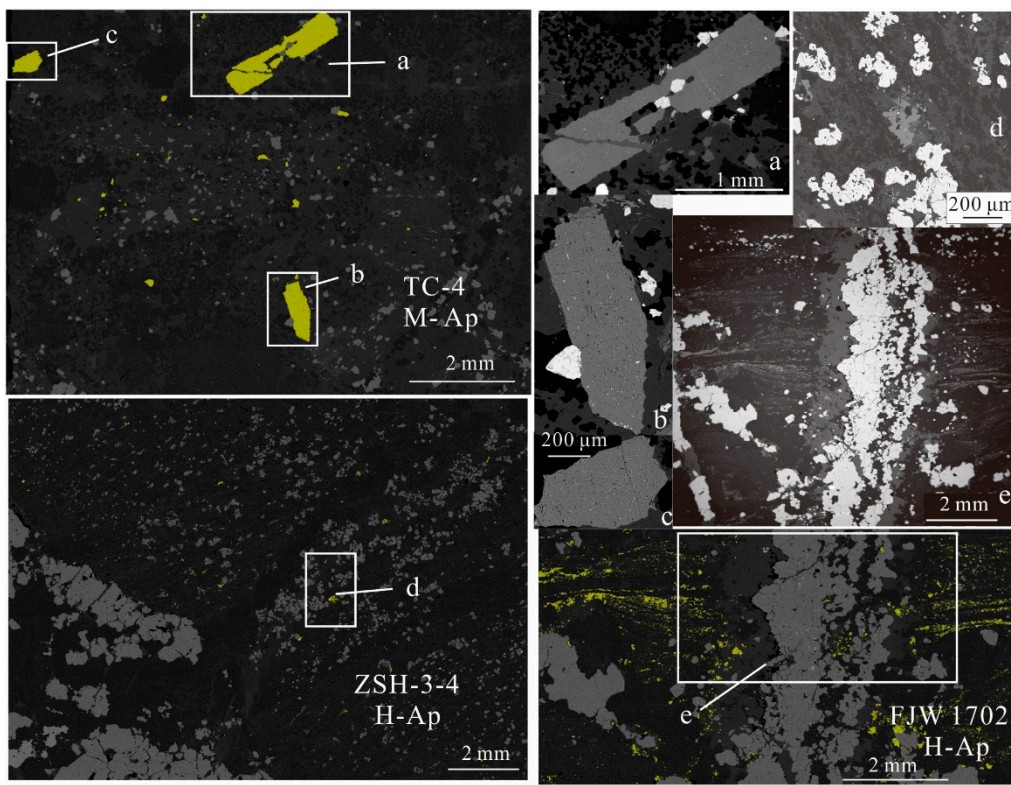

**Figure 4.** Different stages of fluid filling in the Dexing samples and TIMA mapping of the apatite. Apatite grains a, b and c are considered to form in the magmatic process from Tongchang open pit (TC-4); d, e are considered to form in the hydrothermal process from Fujiawu wall rock (FJW-1702) (M: magmatic; H: hydrothermal).

### 4.2. SHRIMP Zircon U-Pb Age Dating

The separated zircon grains, together with zircon standards Plešovice and M257, were mounted and polished. Transmitted and reflected light images as well as CL images of all zircons are used to reveal their internal structures and select potential analysis spots. The mount was vacuum-coated with high purity gold prior to SHRIMP analysis.

Zircon geochronology was conducted using the SHRIMP II ion microprobe at the Beijing SHRIMP Center, Chinese Academy of Geological Sciences. Detailed instrument description and analytical procedure can be found in Williams et al. [55]. The intensity of the primary $O_2^-$ ion beam was ~3nA, while spot sizes ranged from 25 to 30 μm. Zircon standard Plešovice ($^{206}$Pb/$^{238}$U age = 337 Ma, Sláma et al. [56]) was used to calibrate Pb/U ratios, and zircon standard M257 (U = 840 ppm; Th/U~0.27; $^{206}$Pb/$^{238}$U age = 562 Ma) [57] was employed to calibrate U and Th concentrations. $^{204}$Pb was applied for the common lead correction. Data processing was carried out using the Squid and Isoplot programs [58]. Uncertainties in individual analyses are reported at 1σ level, whereas the Concordia U/Pb ages are quoted with 95% confidence interval.

### 4.3. Zircon Hf Isotopes

Zircon Lu-Hf isotopic analyses were performed by using a Nu Plasma II MC-ICP-MS connected to an ASI RESOLution M-50 193 nm laser ablation system at the State Key Laboratory of Continental Dynamics, Northwest University, Xi'an, China. Helium was used as a carrier gas. The spot size was 44 μm, while laser repetition rate was 6 Hz and the energy density applied was 6 J/cm$^2$. Raw count rates for $^{172}$Yb, $^{173}$Yb, $^{175}$Lu, $^{176}$(Hf + Yb + Lu), $^{177}$Hf, $^{178}$Hf, $^{179}$Hf, and $^{180}$Hf were collected simultaneously. The detailed information of these instruments can be found in Bao et al. [59], and the detailed information of analysis strategy and data deduction can be found in Yuan et al. [60]. Zircon standards 91500 and Mudtank were analyzed as external standards to verify the quality of the data during the analysis. The obtained $^{176}$Hf/$^{177}$Hf ratios of the 91500 and Mudtank standards were 0.282307 ± 0.000030 (n = 10, 2σ) and 0.282523 ± 0.000024 (*n* = 10, 2σ), respectively, which are in good agreement with the recommended $^{176}$Hf/$^{177}$Hf ratios within 2σ (0.282311 ± 0.000007 and 0.282520 ± 0.000016, respectively) [60]. The ε$_{Hf}$ values were calculated using a decay constant for $^{176}$Lu of 1.867 × 10$^{-11}$ yr$^{-1}$ [61] and the present-day chondritic ratios of $^{176}$Hf/$^{177}$Hf = 0.282772 and $^{176}$Lu/$^{177}$Hf = 0.0332 [62]. T$_{DM}$ (depleted mantle model ages) were calculated with respect to the depleted mantle with a present-day $^{176}$Hf/$^{177}$Hf ratio of 0.28325 and a $^{176}$Lu/$^{177}$Hf ratio of 0.0384 [63].

### 4.4. Zircon Oxygen Isotopes

Zircon oxygen isotopes were measured using the same SHRIMP II. The $^{133}$Cs$^+$ primary ion beam was accelerated at 10 kV, with an intensity of ~2 nA (Gaussian mode with a primary beam aperture of 200 μm to reduce aberrations) and rastered over a 10 μm area. The spot is about 20 μm in diameter (10 μm beam diameter plus 10 μm raster). The intensity of $^{16}$O was typically 1 × 10$^9$ cps. Oxygen isotopes were measured in multi-collector mode using two off-axis Faraday cups. One analysis takes ~4 min, consisting of pre-sputtering (~120 s), automatic beam centering (~60 s), and integration of oxygen isotopes (10 cycles × 4 s, total 40 s). Uncertainties on individual analyses are reported at 1σ level. The internal precision of a single analysis is better than 0.2‰ for $^{18}$O/$^{16}$O ratio (1σ). δ$^{18}$O values are standardized to VSMOW and reported in standard permil notation.

Ten measurements of the TEMORA 2 zircon standard during the course of this study yielded a weighted mean of δ$^{18}$O = 8.14 ± 0.2‰ (95% confidence level), which is consistent within errors with the reported value of 8.2‰ [64].

### 4.5. Apatite Major and Trace Elements, Sr Isotopes

All of the apatite grains from six samples were selected for EPMA and LA-ICPMS in the State Key Laboratory of Continental Dynamics, Northwest university, Xi'an, China. Apatite grains were carefully observed by microscopy and Raman laser to confirm the mineral inclusion, and some apatite grains were examined using scanning electron microscopy (SEM) and TESCAN (TIMA) area scanning to reveal mineral distribution in thin section.

Major element contents in apatite were determined by electron microprobe analyzer (EMPA). The instrument type JXA-8230, an accelerating voltage of 15 kV, and a beam current of 20 nA were used for all analyses. The spot size of the electron microprobe was about 1 um. The detection limit for Ca was 0.05 wt % with a precision of 0.2% (RSD). A range of natural and synthetic standards were used, and the PAP matrix correction routine has been applied.

Trace elements have been analyzed by the 193 nm ArF laser ablation system coupled with the Agilent 7700x ICP-MS. Detailed descriptions of instrumentation, analytical and calibration procedures, and comparisons with other types of data, are provided by Liu et al. [65]. All the grains' trace elements have been measured at a constant energy 65 mJ, 6 Hz frequency, and ~44 μm spot diameters. Ablated material was transported from the cell to the ICPMS by He carrier gas. A typical analysis consists of 30 s as baseline, 50 s sample signal, and 30 s washout time. Three standards inserted every 8–10 grains are used to correct for any instrumental drift with NIST610, BCR-2, and GSE-1. About 45 trace elements

were obtained through calibration using the NIST 610 glass as the external standard, and Ca analyzed by EMPA as an internal standard.

In situ Sr isotope analyses of apatite were carried out using Nu plasma II MC-ICPMS at the School of Earth and Space Sciences, Peking University. An ArF excimer laser ablation system of Geolas HD (193 nm) was used with 60 μm spot diameter. The samples were ablated for 40 s at a repetition rate of 5 Hz and energy density of 10 J/cm$^2$. The helium gas carrying ablated samples, passing through the "wire" signal smoothing device [66], was mixed with argon gas before entering the MC-ICPMS. AP-1 and SDG were used as internal standards to correct for instrument drift and the value of $^{87}Sr/^{86}Sr$ = 0.71138 ± 0.00018, 0.70300 ± 0.00006 (2SE, $n$ = 16) was obtained, which is consistent with the suggested value of $^{87}Sr/^{86}Sr$ = 0.709179 ± 54 and 0.70298 ± 0.00006 [67].

## 5. Results

### 5.1. Zircon SHRIMP U-Pb Age

Samples TC-W-1, TC-12, and TC-13 are all granitic porphyries from the Tongchang open pit, and the zircons from three samples are mostly euhedral to subhedral and 300–500 μm in length, with length to width ratios of ~3:1. Most zircons are transparent and light brown in color, and most crystals show irregular concentric zoning under CL (Figure 4). U and Th contents vary from 221 to 902 ppm and from 85 to 853 ppm, respectively. Th/U ratios are ~0.3–1.0. All analyses have concordant U-Pb ages within analytical errors, yielding SHRIMP weighted mean U-Pb ages of 168.9 ± 1.2 Ma, 168.0 ± 1.0 Ma, and 172.8 ± 1.3 Ma from the Tongchang porphyry TC-W-1, TC-12, and TC-13, respectively (Figure 3; Table 2, Table S1).

**Table 2.** Summary of SHRIMP U-Pb ages and Hf-O isotopes of zircon from the Dexing porphyries.

| Sample Name | $^{206}Pb/^{238}U$ | 1SE | $\delta^{18}O$ (‰) (2SD) | $^{176}Hf/^{177}Hf$ (2SD) | $\varepsilon_{Hf}(t)$(2SD) |
|---|---|---|---|---|---|
| TC-W-1 | 168.9 | 1.2 | 5.4 ± 0.2‰ | 0.282828 ± 0.000044 | 5.2 ± 1.5 |
| TC-12 | 168 | 1.0 | 5.6 ± 0.2‰ | 0.282816 ± 0.000043 | 5.6 ± 1.6 |
| TC-13 | 172.8 | 1.3 | 5.5 ± 0.2‰ | 0.282840 ± 0.000044 | 4.6 ± 1.5 |
| ZSH-3-4-12 | 830 | 7 | 7.4 ± 0.5‰ | 0.282471 ± 0.000044 | −6.0 ± 1.6 |
| FJW1702-16 | 829 | 8 | 8.1 ± 0.4‰ | 0.282262 ± 0.000046 | −8.6 ± 1.7 |
| FJW-13 | 899 | 12 | 7.4 ± 0.4‰ | 0.282222 ± 0.000049 | −10.9 ± 1.7 |

Samples FJW1702-16, FJW-13, and ZSH-3-4-12 are from the Shuangqiaoshan Group wall rocks from the Fujiawu and the Zhushahong drill cores, and the zircons from these samples are mostly subhedral to cracked, 50–200 μm in length, with length to width ratios of ~2:1. Most zircons are brown to dark brown in color, and most crystals show feeble zoning under CL (Figure 3). U and Th contents vary from 28 to 1031 ppm and from 30 to 1212 ppm, respectively (Table S1), with Th/U ratios from ~0.1–1.5. The zircons of samples FJW1702-16, FJW-13, and ZSH-3-4-12 yield weighted U-Pb ages of 829 ± 8 Ma, 830 ± 7 Ma, and 899 ± 12 Ma, respectively (Figure 3; Table 2).

### 5.2. Zircon Hf isotope

In situ Hf isotopic analysis of 119 zircons from six samples on U-Pb-dated zircons was conducted (Table S1). The $^{176}Hf/^{177}Hf$ of zircons in the porphyry samples TC-W-1, TC-12, and TC-13 are 0.282828 ± 0.000044 (2SD), 0.282816 ± 0.000043 (2SD), and 0.282840 ± 0.000044 (2SD), respectively. The $\varepsilon Hf(t)$ values of zircons TC-W-1, TC-12, and TC-13 are 5.2 ± 1.5 (2SE), 5.6 ± 1.6 (2SE), and 4.6 ± 1.5 (2SE), respectively. Zircon$^{176}Hf/^{177}Hf$ values in wall rock samples ZSH-3-4-12, FJW1702-16, and FJW-13 are 0.282471 ± 0.000044 (2SD), 0.282262 ± 0.000046 (2SD), and 0.282222 ± 0.000049 (2SD), and the $\varepsilon Hf(t)$ values calculated by 170 Ma yielded −6.0 ± 1.6 (2SE), −8.6 ± 1.7 (2SE), and −10.9 ± 1.7 (2SE) (Figure 5a,b; Table 2), respectively.

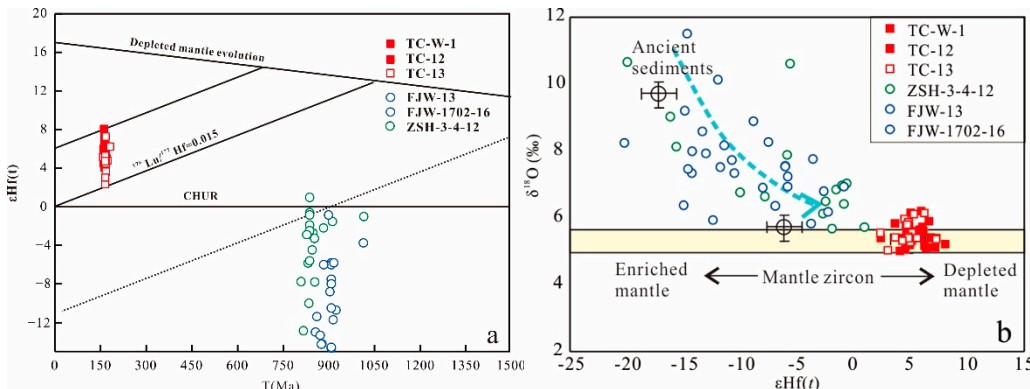

**Figure 5.** (**a**) Zircon Hf isotope composition of the Dexing porphyry copper deposit indicating different sources from the wall rocks; (**b**) plot of in situ zircon $\varepsilon$Hf($t$) vs. $\delta^{18}$O values for the Dexing porphyry and wall rocks.

*5.3. Zircon O Isotope*

In situ oxygen isotopic analysis on 107 zircon grains from six samples are reported in Table S1 and summarized in Table 2. The measured zircon $\delta^{18}$O analyses from the porphyry samples TC-W-1, TC-12, and TC-13 yield average values of 5.4 ± 0.2‰ (2SD), 5.6 ± 0.2‰ (2SD), and 5.5 ± 0.2‰ (2SD), respectively. The $\delta^{18}$O analyses of the wall rock samples ZSH-3-4-12, FJW1702-16, and FJW-13 yield values of 7.4 ± 0.46‰ (2SD), 8.1 ± 0.4‰ (2SD), and 7.4 ± 0.4‰ (2SD), respectively. In general, the distribution range of $\delta^{18}$O in porphyry is relatively concentrated and the ratio is relatively uniform, with a weighted average of 5.5 ± 0.2‰ (2SD), which is different from the distribution of $\delta^{18}$O in the wall rock of the Shuangqiaoshan Group, with a more dispersed distribution and a weighted average of 7.7 ± 0.4‰ (2SD) (Figure 6a).

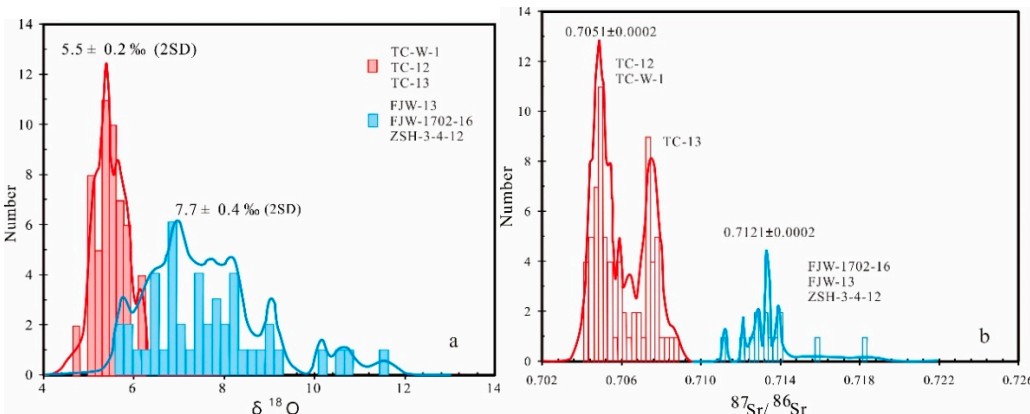

**Figure 6.** The In-situ $\delta^{18}$O and $^{87}$Sr/$^{86}$Sr isotopes characteristics from Dexing porphyry and wall rocks. (**a**) Zircon oxygen isotope characteristics; (**b**) Apatite $^{87}$Sr/$^{86}$Sr composition.

*5.4. Apatite Elemental Compositions*

A total of 76 grains of apatite from three Dexing porphyries and three Shuangqiaoshan wall rock samples have different external features. Under the microscope, the apatite grains from three porphyries have the similar characteristics of complete crystallization, large particles (200~500 μm), colorless or transparent, and no hole in the surface. In contrast, the Shuangqiaoshan wall rock apatite grains are amorphous, with green or lighter green color, small particles (100~300 μm), loose structure, and holes in the surface. The inclusions in all grains vary from finely distributed micro- to nanometer scale inclusions to irregular, large ones (~tens of μm) that are randomly distributed. Sometimes, hematite, pyrite, zircon, and

dolomite inclusions are observed in some apatite as well (e.g., TC-W-1). The major element contents in apatite are summarized in Table 3.

**Table 3.** Summary of the apatite major element contents of EPMA from six samples.

| Sample Name | TC-W-1 | TC-12 | TC-13 | FJW1702-16 | FJW-3 | ZSH-3-4-12 |
|---|---|---|---|---|---|---|
| CaO | 54.09 | 53.81 | 54.11 | 54.73 | 54.80 | 54.54 |
| $P_2O_5$ | 41.67 | 41.38 | 42.2 | 42.36 | 42.39 | 42.35 |
| F | 1.38 | 1.55 | 1.99 | 2.42 | 2.61 | 2.42 |
| Cl | 1.84 | 1.71 | 1.34 | 0.67 | 0.35 | 0.55 |
| FeO | 0.13 | 0.1 | 0.33 | 0.14 | 0.19 | 0.34 |
| MnO | 0.13 | 0.11 | 0.23 | 0.24 | 0.14 | 0.21 |
| $TiO_2$ | 0.02 | 0.02 | 0.03 | 0.04 | 0.04 | 0.04 |
| $SiO_2$ | 0.16 | 0.18 | 0.01 | 0 | 0.05 | 0.04 |
| MgO | 0.01 | 0.02 | 0.07 | 0.1 | 0.06 | 0.07 |
| $Na_2O$ | 0.09 | 0.08 | 0.11 | 0.09 | 0.08 | 0.08 |
| $Al_2O_3$ | 0.08 | 0.02 | 0.02 | 0.09 | 0.11 | 0.03 |
| SrO | 0.02 | 0.02 | 0.01 | 0.11 | 0.04 | 0.04 |
| $SO_3$ | 0.11 | 0.1 | 0.1 | 0.05 | 0.16 | 0.13 |
| Total (%) | 99.73 | 99.09 | 100.55 | 101.04 | 101.01 | 100.85 |

F and Cl contents in six apatite samples have the distinctive ranges from the three ore districts: the average F content in Tongchang samples (TC-W-1, TC-12) was 1.46 wt%, the average Cl content was 1.77 wt%, with a F/Cl ratio of 1.27, which was different from the wall rocks (ZSH-3-4-12, FJW1702-16, and FJW-13) that yield an average F content of 2.53 wt%, an average Cl content of 0.50 wt%, and a F/Cl ratio of 2.16 (Table 3).

Results showed that TC-W-1 and TC-12 have higher average REE (8059 ppm) contents compared to samples TC-13 (5218 ppm), ZSH-3-4-12 (4432 ppm), FJW1702-16 (2028 ppm), and FJW-13 (2730 ppm). All the REE in apatite overall showed a "right pattern", with LREE enrichment and HREE depletion, and the LREE/HREE has the obvious fractionation reaching 17.5, higher than TC-13, ZSH-3-4-12, FJW1702-16, and FJW-13, reaching only 5.6, 5.1, 5.8, and 3.0, respectively. Tongchang porphyries have relatively negative δEu characteristics compared to those of Zhuashahong and Fujiawu wall rocks, where porphyries (TC-W-1, TC-12, and TC-13) yield δEu average values of 0.54, 0.45, and 0.54, respectively, in contrast to wall rock apatites (FJW-1702-16, FJW-13, and ZSH-3-4-12) yielding average values of 1.38, 1.49, and 1.22, respectively (Table 3).

Porphyry samples TC-W-1 and TC-12 have higher average As contents (66.1 ppm) than those of samples TC-13, ZSH-3-4-12, FJW1702-16, and FJW-13 (av. 9.2 ppm). Meanwhile, TC-W-1 and TC-12 have lower contents in Mg (195 ppm), Mn (815 ppm), and Fe (1137 ppm) than TC-13, ZSH-3-4-12, FJW1702-16, and FJW-13, that yield higher contents of Mg (656 ppm), Mn (1861 ppm), and Fe (2709 ppm) (Table 4).

TC-W-1 and TC-12 have higher average Ce contents (av. 3838 and 3378 ppm), different from those of samples TC-13, ZSH-3-4-12, FJW1702-16, and FJW-13 (av. 1786, 1375, 539, and 799 ppm, respectively). Meanwhile, TC-W-1, TC-12, and TC-13 also have higher contents in Pb, Th, and U contents than ZSH-3-4-12, FJW1702-16, and FJW-13 (Tables 4 and S2). The contents of Th and U in porphyry apatite are significantly higher than those in wall rocks (Table 3). The contents of Th and U in TC-W-1, TC-12, and TC-13 apatite are 53.7 and 41.6 ppm, respectively, which are different from those in wall rock from the Fujiawu and the Zhushahong mines, of 22.1 and 21.1 ppm.

**Table 4.** Summary of the apatite trace elements, F/Cl, LREE, and Sr isotope data.

| Sample Name | TC-W-1 | TC-12 | TC-13 | FJW1702-16 | FJW-3 | ZSH-3-4-12 |
|---|---|---|---|---|---|---|
| As | 55.5 | 77.4 | 6.9 | 13.9 | 4.3 | 15.6 |
| Fe | 1230 | 1045 | 3517 | 1268 | 2089 | 3962 |
| Mn | 845 | 811 | 1665 | 1041 | 1567 | 1419 |
| Sr | 354 | 464 | 473 | 602 | 1824 | 423 |
| Rb | / | 1.1 | 0.9 | 2.9 | 2.4 | 2.9 |
| Th | 46.9 | 78.6 | 36.2 | 6.5 | 21.7 | 37.0 |
| U | 33.9 | 56.2 | 35.1 | 5.7 | 24.3 | 31.2 |
| ΣREE | 7932 | 8195 | 5218 | 2028 | 2729 | 4432 |
| Sr/Y | 0.74 | 0.46 | 0.29 | 1.31 | 1.33 | 0.42 |
| LREE | 7638 | 7558 | 4429 | 1732 | 2047 | 3708 |
| HREE | 295 | 636 | 789 | 297 | 683 | 724 |
| LREE/HREE | 34.9 | 15.4 | 6.5 | 6.4 | 3.1 | 6.1 |
| Th/U | 1.37 | 1.64 | 0.96 | 0.92 | 1.06 | 1.14 |
| Ce/Pb | 1424 | 808 | 418 | 156 | 107 | 594 |
| Fe/As | 27.0 | 18.7 | 529 | 212 | 840 | 494 |
| Mn/As | 18.9 | 15.1 | 361 | 170 | 1002 | 254 |
| logfO2 | −11.6 | −11.5 | −13.4 | −12.0 | −13.2 | −12.9 |
| ΔFMQ | 1.84 | 1.92 | 0.04 | 1.26 | 0.23 | 0.58 |
| $^{87}Sr/^{86}Sr$ | 0.7047 | 0.7050 | 0.7077 | 0.7124 | 0.7078 | 0.7172 |
| 2SD | 0.0007 | 0.0004 | 0.0004 | 0.0003 | 0.0004 | 0.0016 |

In the Ce/Pb vs. Th/U diagram (Figure 7a), the Ce/Pb ratios of Tongchang porphyry apatite are much higher than those of Fujiawu and Zhushahong apatites in wall rocks. The Ce/Pb and Th/U ratios of Tongchang porphyry apatite are generally higher, yielding average ratios of 728 and 1.3, respectively, reflecting the strong fluid activity during the formation of ore-bearing porphyry magma. In contrast, the Ce/Pb and Th/U ratios are generally lower in the Zhushahong and the Fujiawu mines (252 and 1.04, respectively), reflecting the weak fluid activity during the formation of the ore-barren wall rock and the insignificant magmatic differentiation, which is closely related to the strength of the fluid action in the magmatic system.

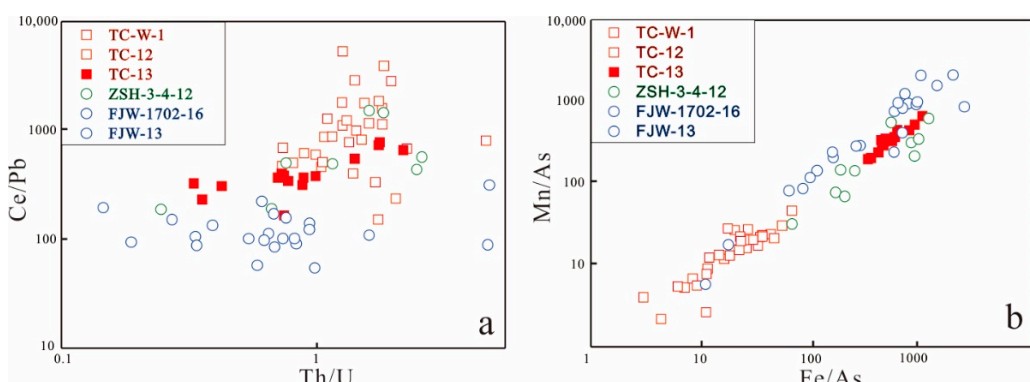

**Figure 7.** The apatite trace element ratios diagram in the Dexing deposit, (**a**) Ce/Pb vs. Th/U; (**b**) Mn/As vs. Fe/As.

In the Mn/As vs. Fe/As diagram (Figure 7b), the range of TC-W-1 and TC-12 apatite is much narrower than that of TC-13, ZSH-3-4-12, FJW1702-16, and FJW-13 apatite, which

have relatively broad variation and show a clear linear relationship. The TC-W-1 and TC-12 magma ratio of porphyry apatite is much higher than that of altered apatite.

### 5.5. Apatite Sr and Sr Isotope

All grains have Sr contents higher than 200 ppm (av. 690 ppm, Table 4). Tongchang and Zhushahong apatites have relatively low Sr contents (av. 354~602 ppm, up to 836 ppm), distinctive from the Fujiawu apatite (FJW-13), with high Sr contents (up to 1824 ppm). The Y contents of apatite in porphyry and wall rock are 1257 ppm and 1130 ppm, respectively (Table 4; Table S2).

TC-W-1 and TC-12 have the lowest $^{87}Sr/^{86}Sr$ ratios (0.70481 ± 0.00016), different from those of TC-13 and FJW-13 (0.70772 ± 0.00040), ZSH-3-4-12 (0.71716 ± 0.03104), and FJW1702-16 (0.71173 ± 0.00016).

## 6. Discussion

### 6.1. Different Oxygen Fugacity of Parental Magmas and Wall Rocks

Oxygen fugacity (fO$_2$) is variable in magmatic and magmatic-hydrothermal systems, controlling the behavior of redox-sensitive elements [68]. Some redox-sensitive elements (e.g., Fe, S, V, Cr, or Ce) with different valence in silicate glasses and accessory mineral phases can be used to calculate fO$_2$. Particularly, both zircon and apatite are used to constrain the variation of oxygen fugacity associated with magmatic and hydrothermal processes.

#### 6.1.1. Zircon Oxybarometer

Zircon Ce/Ce$^*$ and (Eu/Eu*)$_N$ as the oxybarometer can provide the information about the oxidation and hydration state of the parental magmas [4,5,16–19]. Ce/Ce$^*$ and Ce$^{4+}$/Ce$^{3+}$ ratios can demonstrate direct spectroscopic measurements or estimation of Ce$^*$ by interpolation between La and Pr or by extrapolation from middle REEs using the lattice-strain method of Blundy et al. [69]. They are very sensitive to the temperature of zircon crystallization. Notably, Eu$^{3+}$/Eu$^{2+}$ and Eu/Eu* in zircon is not as sensitive to oxidation as Ce$^{4+}$/Ce$^{3+}$ and Ce/Ce* because of the influence of plagioclase crystallization precipitated from the melt prior to zircon crystallization.

The Ce/Ce$^*$ and (Eu/Eu*)$_N$ ratios of zircons in the Tongchang porphyries are 126~2022 (av. 719) and 0.51~0.93 (av. 0.74), respectively. These values are equal to the hematite-magnetite buffer based on the hematite-magnetite overgrowth identified under the microscope, but higher than those from the Chuquicamata-El Abar samples [4,37]. Zhang et al. [7] also obtained the Ce/Ce* ratios ~570 and 580 in porphyries of the Tongchang and Fujiawu mines, respectively. In contrast, the Ce/Ce* and (Eu/Eu*)$_N$ ratios in the Zhushahong and the Fujiawu wall rocks are 4.6~757 (av. 154) and 0.03~1.00 (av. 0.39), respectively.

In samples TC-W-1 and TC-12, zircon provides the higher Ce$^{4+}$/Ce$^{3+}$ ratios. The magmatic process that reached highest oxygen fugacity, in turn, is FJW-1702-16 to ZSH-3-4-12. FJW-13 and TC-13, and TC-13 yielded the lowest ΔFMQ values (Figure 8). We propose that TC-W-1 and TC-12 in the Tongchang mine may have experienced the relative higher oxygen fugacity (ΔFMQ + 1.5) reached in the magmatic process, which also obtained the value refined to ΔFMQ + 1.5 using zircon Ce$^{4+}$/Ce$^{3+}$.

Loucks et al. [19] also established a novel method for determining the oxidation state of a magma as zircon crystallized using Ce, U, and Ti contents in zircon, providing the fO$_2$ equation: $\log\ fO_{2(sample)} - \log\ fO_{2(FMQ)} = 3.99_8\ (\pm 0.12_4) \times \log\left[\left(Ce/\sqrt{(U_i \times Ti)^z}\right] + 2.28_4\ (\pm 0.10_1)\right.$. Samples TC-W-1 and TC-12 from Tongchang yield the highest ΔFMQ values (Table 4) of 1.35 in TC-W-1 and 1.56 in TC-12, consistent with the results of previous work (ΔFMQ 0.7~1.9, Zhang et al. [7]). The wall rocks in the Fujiawu (FJW-1702-16 and ZSH-3-4-12) yield ΔFMQ vales –0.25 to 0.27 (Table 4). Based on the Ce/Ce* and ΔFMQ values, we propose that samples TC-W-1 and TC-12 from Tongchang obtain the highest ΔFMQ values, while the Fujiawu and Zhushahong provide give a relatively lower oxygen fugacity.

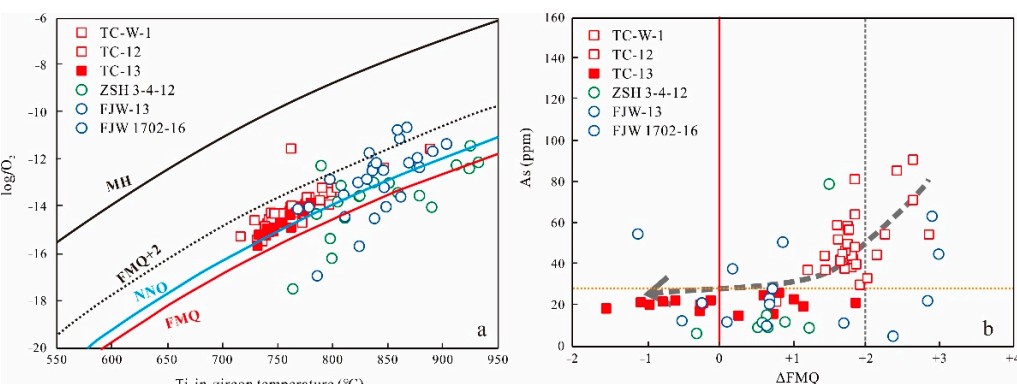

**Figure 8.** Diagrams of (**a**) Ti-in-zircon temperatures (°C) vs. logfO$_2$ and (**b**) apatite As vs. ΔFMQ, the ΔFMQ calculated by apatite Mn contents.

### 6.1.2. Sulfur/Mn-in-Apatite Oxybarometer

Apatite is a nearly ubiquitous accessory mineral containing volatiles and redox-sensitive elements such as Mn and S [70–72]. Manganese dominantly has two valence states (Mn$^{2+}$ and Mn$^{3+}$) in magmas. Generally, the Mn content in apatite increases with decreasing magma fO$_2$ because Mn$^{2+}$ is more compatible with Ca$^{2+}$ sites [73]. Accordingly, Miles [13,23] proposed a Mn-in-apatite oxybarometer. Manganese varies linearly and negatively with log fO$_2$. An equation of logfO$_2$ = −0.0022(±0.003)Mn (ppm) − 9.75(±0.46) has been set up based on the Griffell granitic pluton apatite Mn contents in south Greenland; yet, whether Mn in apatite is redox-sensitive is still disputed. Reflecting the hydrothermal process, ZSH-3-4-12, FJW-13, and FJW-1702-16 of the Zhushahong and Fujiawu mine also present relatively low magma oxygen fugacity—about ΔFMQ-0.25~+1.00—with weak propylitic to phyllic alteration. Meanwhile, it was thought that the ore-forming system of Tongchang mine had recorded relatively high oxygen fugacity, different from the wall rock with low oxygen fugacity.

Apatite from Tongchang (TC-W-1 and TC-12) exhibits the highest logfO$_2$ and ΔFMQ values (1.97 and 1.84, respectively), which is the best Cu mineralization redox condition reached in the hematite-magnetite buffer, consistent with previous work [37]. In contrast, the wall rock of the Fujiawu and the Zhushahong mines, exhibiting relatively low oxygen fugacity, do not represent ideal mineralization conditions. Particularly, the Fujiawu mine samples FJW-1702-16 and FJW-13 yield logfO$_2$ and ΔFMQ vales of −12.0 and 1.41, and −12.9 and 0.55, respectively. In the Zhushahong mine sample, ZSH-3-4-12, the logfO$_2$ and ΔFMQ values obtained are −12.7 and 0.68, respectively.

SO$_3$ content in apatite has shown the potential to serve as a powerful sulfo- and oxy-barometer for a broad range of natural systems [20]. In the Dexing porphyries, TC-W-1 and TC-12 have high sulfur contents, from 0.02~0.19 (av. 0.11). The wall rock samples with mineralization (ZSH-3-4-12, FJW-13, and FJW-1702-16) also have relatively variable sulfur contents, from 0.01~0.55 (av. 0.18), which is consistent with the ore-bearing porphyry apatite (Figure 9a; SO$_3$ > 0.1%; [74]).

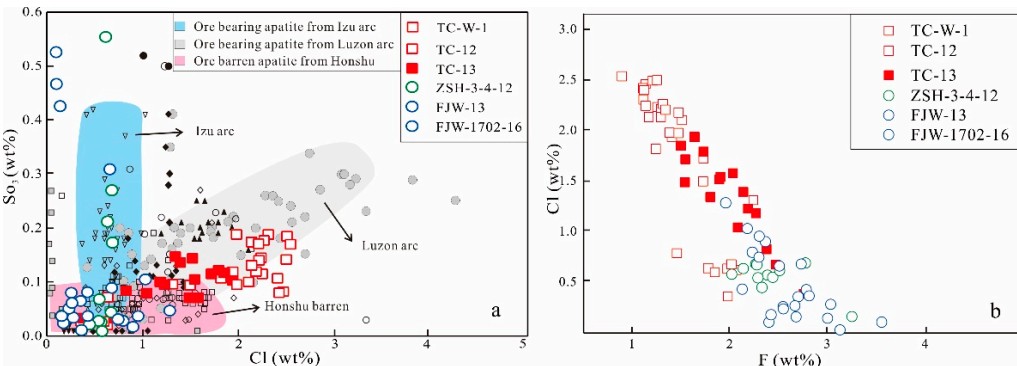

**Figure 9.** The Dexing porphyries and wall rock apatite trace element ratios. (**a**) SO₃ vs. Cl distribution in apatite. The Luzon arc from the Philippines and the Izu Bonin arc are related with deposits, and are distinct from Honshu from northeastern Japan that is barren. Modified from Imai et al. [75]; (**b**) F vs. Cl contents' distribution in apatite from the Dexing deposit.

This observation has potential metallogenic significance; i.e., at higher $fO_2$, sulfur is present in magmas as sulfate ($S^{6+}$), whereas in more reduced magmas (below the fayalite magnetite quartz (FMQ) redox buffer) sulfide ($S^{2-}$) predominates [74], which saturates more easily to form an immiscible sulfide melt strongly partitioning Cu (e.g., Patten et al. [76]). This process can lead to the loss of both sulfur and metals from the melts, thereby potentially reducing their ore-forming potential. Taken together, these observations have been used as evidence that: (1) porphyry magmas are highly oxidized, similar to typical arc magmas (>ΔFMQ + 2; [3]); (2) that $fO_2$ increases through successive int rusions leading up to mineralization, and therefore may be a controlling factor in ore formation; and/or (3) wall rocks in Fujiawu and Zhushahong are all relatively reduced, precluding Neoproterozoic strata as the source of parental magmas. Zhang et al. [7] also present low oxygen fugacity values from these Neoproterozoic-inherited zircons (FMQ − 2.4~0.7)— much lower than those of middle Jurassic metallogenic magmatic rocks (FMQ + 0.7~+1.9) in one zircon. Therefore, the Neoproterozoic arc cannot generate such oxidized magmas.

*6.2. Evolution of Ore-Forming Systems*

Volatiles (in particular $F^-$, $Cl^-$, $OH^-$, and $CO_3^{2-}$) play an important role in magmatic-hydrothermal ore-forming processes in porphyry systems, because they affect physicochemical properties such as magma viscosity and density [9,77,78], timing of fluid exsolution in the magma chamber, and partitioning of trace elements between silicate melt and aqueous fluid. Cl is one strongly incompatible element that can enter the fluid easier than F in magma under fluid dissolution. In the F vs. Cl diagram (Figure 9b), samples TC-W-1 and TC-12 have higher Cl contents, different from ZSH-3-4-12, FJW1702-16, and FJW-13, with higher F contents, indicating that Cl plays a more important role in the metal migration process than F, and excess F entry into the hydrothermal apatite is easier than that of Cl during the mineralization process. In the process of subduction, Cl tends to leave with the dehydration of the subducted slab into the arc and back arc basin, or in the subduction zone and mantle fluid enrichment [79,80].

The total of LREE contents is the main difference between hydrothermal apatite and magmatic apatite, where TC-W-1 and TC-12 have higher LREE than Zhushahong samples, and Fujiawu samples have even lower LREE contents (Figure 10a). In the LREE vs. Sr/Y diagrams, all of the Tongchang apatites fall in the mafic I-type granitoids and mafic igneous rocks field (IM), and the Zhushahong and Fujiawu apatites mostly override fields of IM and low-to-medium-grade metamorphism (LM) and high-grade metamorphism (HM), indicating that these grains underwent different stages of alteration based on the wall rocks changed by metamorphism (Figure 10b).

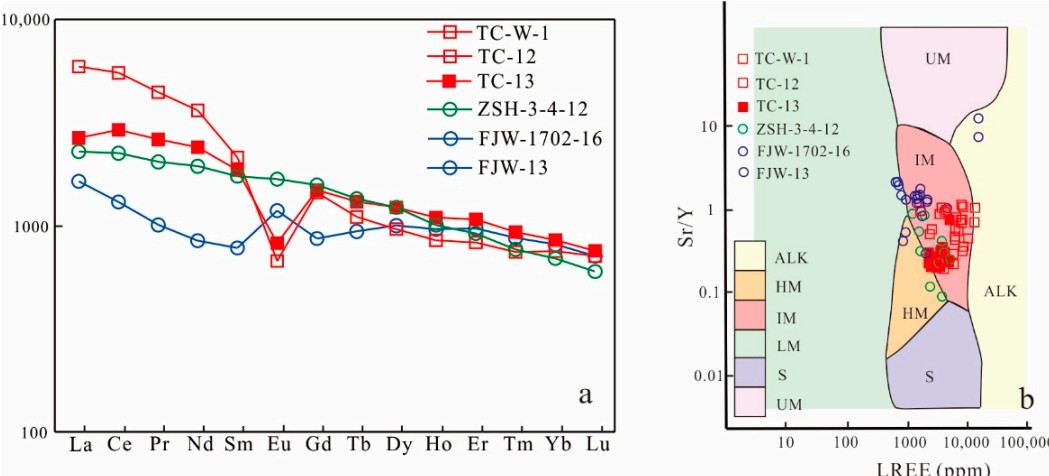

**Figure 10.** (**a**). The REE patterns of apatite; (**b**). Sr/Y vs. ΣLREE diagram, modified from O'Sullivan [81]. Abbreviations for groups: ALK = alkali-rich igneous rocks; IM = mafic I-type granitoids and mafic igneous rocks; LM = low- and medium-grade metamorphic and metasomatic; HM = partial-melts/leucosomes/high-grade metamorphic; S = S-type granitoids and high aluminum saturation index (ASI) "felsic" I-type; UM = ultramafic rocks including carbonatites, lherzolites, and pyroxenites.

Thus, we summarize the discrimination between hydrothermal and magmatic apatite as the following: (1) Hydrothermal apatite has higher $SiO_2$, MnO, $SO_3$, and F contents, possesses trace element such as As, Th, U, LREE, and ΣREE, and possesses a different Eu anomaly, compared to magmatic apatite. (2) Based on the chemical characteristics—such as ion charge, ionic radius, and element compatibility—of different elements, they also provide different distributions of elemental ratios, such as F/Cl, Ce/Pb, Th/U, Sr/Y and As/Fe. (3) The increasing Sr isotopes and Mn/As ratios of the samples from the Tongchang mine suggest minor contribution from wall rocks during magma evolution.

*6.3. Endowment in Cu Mineralization from Reworking of Neoproterozoic Strata*

Previous studies have suggested that the high oxygen fugacity of the Dexing porphyry copper deposit was inherited from the Neoproterozoic island arc rather than the subduction of the Paleo-Pacific plate during the Jurassic [35], and the Cu metal may have originated from the thickened lower crust during remelting of the Neoproterozoic arc. The Neoproterozoic Shuangqiaoshan Group consists of a series of metamorphic volcanic-sedimentary rocks and related epizonal metamorphic rocks [32]. The basement strata in the Dexing area are composed of the Shuangqiaoshan and Qigong groups. The Shuangqiaoshan Group was subdivided, from bottom to top, into the Hengyong, Jilin, Anlelin, and Xiushui formations [82]. The Hengyong Formation occurs strictly in the Dexing area as the wall rock to the Dexing Cu (-Mo-Au) deposits that formed at ca. 830 Ma, named Jiudu Formation in the literature.

The zircon and apatite in the porphyry have U-Pb ages, and Hf-O and Sr isotopic compositions that differ from those of the wall rocks. The Dexing porphyry copper deposit yields an age range from 167 to 173 Ma. In contrast, the Shuangqiaoshan Group wall rocks have zircons with U-Pb ages of $830 \pm 10$ Ma, $830 \pm 10$ Ma, and $899 \pm 11$ Ma (Figure 3). These ages of wall rocks are relatively complex and represent the age of the Hengyong and Jilin formations.

The $^{176}Hf/^{177}Hf$ and εHf(*t*) of zircons in porphyry samples is 0.282816 to 0.282840, and 4.9 to 8.5, respectively. These values are very different from those of the wall rock from the Zhushahong and the Fujiawu drill cores (0.282222 to 0.282471, and −7 to 16, respectively). In general, the $^{176}Hf/^{177}Hf$ ratios of the porphyry are relatively uniform, which is different from that of wall rock. The measured zircon $\delta^{18}O$ average values from three porphyry

samples range from 5.4 to 5.6, presently different from three wall rock samples (7.4 to 8.1). Meanwhile, oxygen isotope characteristics show that the $\delta^{18}O$ distribution range in porphyry provide a relatively narrow range, with a weighted average value of $\delta^{18}O$ of 5.5, which is different from wall rocks (weighted average value of $\delta^{18}O$ of 7.7). The $^{87}Sr/^{86}Sr$ ratio of apatite of the porphyry is obviously different from those of wall rocks as well. The $^{87}Sr/^{86}Sr$ ratio of the porphyry has peak distribution of $0.7051 \pm 0.0002$, whereas the $^{87}Sr/^{86}Sr$ ratio of apatite in the wall rocks has a peak distribution of $0.7121 \pm 0.0002$. The wall rocks have much higher $^{87}Sr/^{86}Sr$ ratios than those of the porphyries.

Previous studies found a large number of inherited zircons in the Dexing copper deposit (642-842 Ma [7]). Based on the method of Smythe, Brenan et al. [83] to calculate zircon $Ce^{4+}/Ce^{3+}$ as the indicator of oxygen fugacity buffer, Zhang et al. [7] determined that the composition of the source rock of the inherited zircon in the Dexing deposit is similar to the Pingshui andesite volcanic rock in the Shuangxiwu Formation, Neoproterozoic strata exposed in Shaoxing area. Moreover, the Hf-Nd isotopic composition in the Dexing deposit is similar to that of the Neoproterozoic island arc rocks, indicating that it may come from the partial melting of the subducting Paleo-Pacific plate [34–36,84].

### 6.4. Metallogenesis of the Giant Dexing Cu-Deposit

Previous studies on the metallogenic mechanism of the Dexing porphyry copper deposit found that the deposit was formed in the typical intraplate metallogenic environment, with higher oxygen fugacity conditions. The researchers have provided many models for its intracontinental origin, including subduction with the Paleo-Pacific Ocean, melting of thickened lower crust, and remelting of the lower crust. The main dispute is focused on the large-scale effects of either Jurassic Pacific subduction or the subcontinental remelting with subduction. Here, the key of the intraplate setting with high oxygen fugacity porphyry copper mineralization provides more detailed information using zircon and apatite.

Through the in-depth study of accessory minerals, the main factors that can provide direct evidence for the genesis of the deposit are as follows: (1) Based on the abnormalities of high oxygen fugacity indexes $Ce/Ce^*$ and $(Eu/Eu^*)_N$ of zircon, the genesis and sources of the granodiorite porphyry are similar to the highly oxidized Cu-Au deposits in the southern United States. The symbiosis of anhydrite and magnetite-hematite is the same, which thus indicates that the high oxygen fugacity of Dexing porphyry has reached the critical position of hematite-magnetite. (2) In situ Hf-O isotope study of zircon and apatite compositions suggest that the source of ore-forming material and hydrothermal fluid may be dominantly by mantle-derived magma, but different from metasomatized wedge. (3) In terms of the regional structure and tectonic environment, the Dexing porphyry copper deposit is located along the northwest side of the large deep fault in northeastern Jiangxi, with developed fold structures, crisscrossing faults, and densely integrated cracks. Among them, nearly 30 first-level NNE-trending and second-level E-W-trending faults are mainly developed, which are superimposed with multi-stage folds to provide favorable channels and storage space for the upwelling of ore-forming elements and hydrothermal fluids. At the same time, the Dexing area of South China is located on the outer edge of the Pacific subduction zone, where slab subduction provides the most favorable tectonic environment for mineralization.

Based on the evolutionary characteristics and drift history of the Pacific plate, we believe that the formation of the parental magma of Dexing is related to slab melting, which is probably related to the large-scale mineralization associated with subduction of the Paleo-Pacific Ocean during the Middle Jurassic.

### 7. Conclusions

1. The porphyries of the giant Dexing Cu deposit formed in the Jurassic (172–168 Ma), whereas its host rock in the hanging wall was formed in the Neoproterozoic, based on the ages of the volcanics in the Shuangqiaoshan Group (900–830 Ma).

2.　The porphyries display relatively low zircon $\delta^{18}$O and apatite $^{87}$Sr/$^{86}$Sr, but high zircon $\varepsilon$Hf, indicating a mantle source; however, the wall rocks yield characteristics transitional from magmatic to hydrothermal, implying alteration of ore-forming fluids.

3.　The apatites from the porphyries provide relatively high total REE and negative $\delta$Eu, but high Cl and As contents. Combined, the porphyries underwent potassic–silica alteration, implying processes from magmatic to hydrothermal alteration, which is distinctly different from the Shuangqiaoshan Group rocks with hydrothermal fluid from the lower crust.

4.　The Dexing porphyries were derived from partial melting of the subducted Paleo-Pacific plate during the Jurassic rather than from that of the lower crust. High oxygen fugacity and high Cu and Cl contents are the key features for its metallogenesis.

**Supplementary Materials:** The following are available online at https://www.mdpi.com/article/10.3390/min12111464/s1, Table S1: Zircon SHRIMP ages, Hf-O isotopes, Laser U-Pb ages and trace elements from Dexing porphyry copper deposit. Table S2: Apatite major elements, trace elements and Sr isotopes from Dexing porphyry copper deposit.

**Author Contributions:** H.Z. wrote the paper; F.A., X.F. and M.L. designed the experiments and comments; W.S. as the visulization, supervisor, funding acquisition. All authors have read and agreed to the published version of the manuscript.

**Funding:** This work was supported by the National Natural Science Foundation of China (No. 41402064, 42130102), Marine S&T Fund of Shandong Province for Pilot National Laboratory for Marine Science and Technology (Qingdao) (No.2022QNLM050201-4).

**Acknowledgments:** The authors are grateful to two anonymous reviewers for their helpful comments and suggestions that greatly helped to improve an earlier manuscript version. We greatly thank Gao, J. F. from institute of geochemistry Chinese academy of science as the earier manuscript supervision and given comments many times, and we also appreciate for Peng, P., Li, Q. L., Mitchell, R. and Yang, Y. H. from institute of geology and geophysics, Chinese academy of science for suggestion, data measure and check.

**Conflicts of Interest:** I warrant the manuscript represents original work that is not being considered for publication, in whole or in part, in another journal, book, conference proceedings, or government publication with a substantial circulation. I warrant there is not any conflict of interests.

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
