# Peer review of "Metallogenesis of Porphyry Copper Deposit Indicated by In Situ Zircon U-Pb-Hf-O and Apatite Sr Isotopes"

_minerals, doi:10.3390/min12111464_

Round 1

Reviewer 1 Report

The paper under revision contains lots of novel data. In particular, the isotopic data the authors provide ar of high interest.

However, the article does not make the new data justice at all. First, the geological description is absolutely unsufficient, so that the reader has little or no idea about the rocks and mineralization under study. Regarding the data interpretation, I seriously hesitate that the explanations the authors offer are the only possible.

The only thing I can clearly understand is that the granodioritic (adakitic) rocks hosting the porphyry mineralization and the volcanic/metavolcanic succession intruded by the adakitic rocks strongly differ in age, geochemical features and isotopic signature. Well, I recognize that this excludes any genetic relation between the volcanic succession and the adakitic rocks -and this is something. But such a contrast is probably the most common relation between intrusive rocks and their host tocks: any intrusive rock, any country rock. Regarding the (highly debated) issue of the origin of adakite rocks, I cannot see any relevant contribution. There should be some, in view of the many novel data the author provide.

My conclusion is simple: first of all, the authors should pay better attention to their new, highly interesting data; then they should better organize their descriptions, improving field, petrographic and mineralogical descriptions, and finally they should relate descriptions and interpretations. In doing this latter task, they shoud carefully separate them avoiding too simple interpretations. If this would imply enlarging the discussion section, this would be undoubtedly better that ignoring the many complexities of the issue they deal with.

Finally, and in spite the fact that my mother language is NOT English, I appreciate many language mistakes (even elementary), which make the manuscript very difficult to read. My advice is to send any revised manuscript to a native English geologist for translation. I insist in that he/she should be a geologist because some geological terms are not correctly used by the authors. Maybe a number of the deffects I am attributing to other causes could be related to deficient syntax and spelling

Reviewer 2 Report

Dear sir/ma'am

Thank you for submitting your manuscript "Metallogenesis of porphyry copper deposit indicated by in-situ zircon U-Pb-Hf-O and apatite Sr isotopes" to Journal of Minerals, MDPI. It is an interesting and well-done study on the porphyry copper deposits. I suggest generalizing the study; maybe they will reach a general conclusion on the porphyry copper deposits. In the future, I recommend the authors to think about the contains a revised Conceptual, descriptive and numerical models of porphyry copper deposits (PCDs) and their comparison for block design. In addition, I recommend unifying the form of references and the structure of Journal. I encourage you to write next article.

 No particular mistakes or errors, But:

Comment:

1.  Since the results of this paper cannot be generalized yet, please add, case study at the end of the title.

2.  In Sampling: Please explain more about the flotation method; Do you floated via water? Alternatively, Chemical flotation again?

3.   In Sampling (4-1):

 3-1. Please explain more about the drilling holes and their location relative to each other on the geological map.  How did sampling? Sampling method should be mentioned.

3-2. Have you measured the grade distribution relative to the mesh size? What is the degree of liberation of samples? Please, plot the results on a grade recovery curve.

3-3.Please explain more about the flotation method; do you float the sample in water? Alternatively, Chemical flotation agent? If, yes, please refer to the type of agent.

3-4.Have you measured the grade distribution relative to the mesh size? What is the degree of liberation of samples? Please, plot the results on a grade recovery curve.

4.  Please bold your novelty. What is exactly your novelty?

5.  A short description of the tools used.

6.  Conclusions of the investigation, or, very importantly in this case, what was not yet investigated, which properties, modifications and applications have not yet been investigated and why it is necessary to consider them.

We are looking forward to hearing from you as soon as possible.

Best Regards,

Reviewing Team

Author Response

Referee #2 Comments and Suggestions for Authors

C: Thank you for submitting your manuscript "Metallogenesis of porphyry copper deposit indicated by in-situ zircon U-Pb-Hf-O and apatite Sr isotopes" to Journal of Minerals, MDPI. It is an interesting and well-done study on the porphyry copper deposits. I suggest generalizing the study; maybe they will reach a general conclusion on the porphyry copper deposits. In the future, I recommend the authors to think about the contains a revised Conceptual, descriptive and numerical models of porphyry copper deposits (PCDs) and their comparison for block design. In addition, I recommend unifying the form of references and the structure of Journal. I encourage you to write next article.

R: Thank you very much for your recognition as well as important comments and suggestions on our revised manuscript. According to your following specific comments and suggestions, we have made revisions, especially on expression and sampling related to minerals flotation method and petrogenesis of adakitic rocks. We hope this round of revision can meet with the reviewers’ approval.

C.1. Since the results of this paper cannot be generalized yet, please add, case study at the end of the title.

R: Thank you very much for your suggestion. We have add Dexing in the title: Metallogenesis of porphyry copper deposit indicated by in-situ zircon U-Pb-Hf-O and apatite Sr isotopes in Dexing porphyry copper deposit from Southeast China.

C.2. In Sampling: Please explain more about the flotation method; Do you floated via water? Alternatively, Chemical flotation again?

R: Thank you very much for your question, it is very key procedure for drill core samples. We have added the related information about via water flotations.

C:

  1. In Sampling (4-1):

3-1. Please explain more about the drilling holes and their location relative to each other on the geological map.  How did sampling? Sampling method should be mentioned.

R: Thank you very much about the suggestion. We have described sampling method in the revised version in detailed. About 10 cm of fresh wall rocks samples in half-split drill cores from the Zhushahong and Fujiawu mines were carefully sampled. 

3-2. Have you measured the grade distribution relative to the mesh size? What is the degree of liberation of samples? Please, plot the results on a grade recovery curve.

R: Thank you very much about the suggestion about metal degree of samples. All of the porphyries come from the open pit with weak alterations. The wall rocks are all from drill hole and with different degree of alterations and different type veins. Yang (2019) and Zhu (1983) are all present an authoritative metal grade recover curve (Fig. 1).

Fig.1 Density (veins/m) of Cu-Mo-Au-bearing veins in the Tongchang (A) and Fujiawu (B) deposits in the Dexing district from Yang (2019).

3-3.Please explain more about the flotation method; do you float the sample in water? Alternatively, Chemical flotation agent? If, yes, please refer to the type of agent.

Reply: Thank you for your attention to the sample sorting process. Among them, the sample sorting process is traditional process, contain water flotation, then magnetic separation and heavy liquid separation, and finally remove impurities under the binocular lens pick out all of the zircon and apatite at last.

Zircon and apatite are separated from granitic rocks and drillings. The open pit granitic porphyries are probably within 2 to 5kg, while the drilling samples are relatively small within the range of 1-2kg (Fig. 2). Dozens of grains of zircon and apatite was been mount, all of the grains are used for analysis. Among them, the samples called the surrounding rock are mainly clastic rock, the age is mainly for the new Proterozoic and Proterozoic, only these ages of the new Proterozoic got concorida age, others are scatter fall in different stage withnot concordia (Fig. 3).

Fig.2 The sample of FJW1702-11 from Fujiawu drilling. 

Fig.3 the wall rock sample FJW1702-16 from Fujiawu deposit, it present the Shuangqiaoshan Group in this district U-Pb Concordia ages given 829±8 Ma. 

3-4.Have you measured the grade distribution relative to the mesh size? What is the degree of liberation of samples? Please, plot the results on a grade recovery curve.

R: Thank you very much about the samplings again.

  1. Please bold your novelty. What is exactly your novelty?

R: Thank you very much about your suggestion. We think the novelty about this manuscript is mainly focus on the genesis of Dexing porphyries, particular about the magma oxygen fugacity in hydrothermal process. This study uses in situ compositions of zircon and apatite to trace the nature and evolution histories of magma and hydrothermal fluids, constrained that the metal source of Dexing was not the Neoproterozoic arc but subducting slab.

  1. A short description of the tools used.

R: Thank you very much for the suggestion. We have added the description of tools in the new version.

  1. Conclusions of the investigation, or, very importantly in this case, what was not yet investigated, which properties, modifications and applications have not yet been investigated and why it is necessary to consider them.

R: Thank you very much for the comments. We have carefully revised the manuscript to clearly document the objectives, methodologies and new contributions in the new version. Because the porphyry Cu deposits are generally highly altered, whole rock geochemistry can not trace the origin well. But accessary minerals, including apatite and zircon, could be stable during later hydrothermal alterations. Therefore, by using the mineral compositions, we can better constrain the magma sources, and evolution histories of magma and hydrothermal fluids. The new datasets show that the oxygen isotopic compositions are mantle-like and magmas and hydrothermal fluids are highly oxidized, suggesting that parental magmas were derived from partial melting of the subducting paleo-Pacific plate and wall rocks have limited contribution for porphyry Cu deposit.

Yang, Z; Cooke, D. Porphyry copper deposits in China. Society of Economic Geologists Special Publication. 2021, 22, 133-187.

Yours Sincerely,

  Hong Zhang

On behalf of the co-authors

Round 2

Reviewer 1 Report

Regarding the many problems inherent to the adakite genesis, I continue to believe that the article under revision is no panacea. However, I consider that the authors have correctly addressed the points I suggested to change. Accordingly, my recomencation is to publsh this work as it now stands.

Author Response

We sincerely thank Reviewer for constructive review comments, which are helpful for us to improve our manuscript. We have revised all of the detail annotation and check the whole manuscript again. Thank you very much.